# Near-field detection of gate-tunable anisotropic plasmon polaritons in black phosphorus at terahertz frequencies

Eva A. A. Pogna [1,2] ✉, Valentino Pistore [1], Leonardo Viti [1], Lianhe Li [3], A. Giles Davies [3], Edmund H. Linfield [3] & Miriam S. Vitiello [1] ✉

Polaritons in two-dimensional layered crystals offer an effective solution to confine, enhance and manipulate terahertz (THz) frequency electromagnetic waves at the nanoscale. Recently, strong THz field confinement has been achieved in a graphene-insulator-metal structure, exploiting THz plasmon polaritons (PPs) with strongly reduced wavelength ($\lambda_p \approx \lambda_0/66$) compared to the photon wavelength $\lambda_0$. However, graphene PPs propagate isotropically, complicating the directional control of the THz field, which, on the contrary, can be achieved exploiting anisotropic layered crystals, such as orthorhombic black-phosphorus. Here, we detect PPs, at THz frequencies, in hBN-encapsulated black phosphorus field effect transistors through THz near-field photocurrent nanoscopy. The real-space mapping of the thermoelectrical near-field photocurrents reveals deeply sub-wavelength THz PPs ($\lambda_p \approx \lambda_0/76$), with dispersion tunable by electrostatic control of the carrier density. The in-plane anisotropy of the dielectric response results into anisotropic polariton propagation along the armchair and zigzag crystallographic axes of black-phosphorus. The achieved directional subwavelength light confinement makes this material system a versatile platform for sensing and quantum technology based on nonlinear optics.

One of the key challenges for the translation of photonic technologies to the terahertz (THz) frequency range is the long (1 mm –10 μm) free-space wavelength $\lambda_0$ of THz-frequency electromagnetic (e.m.) waves, which, owing to diffraction, prevents the field concentration into length scales much smaller than the wavelength[1].

A promising route to confine THz fields to the nanoscale relies on the exploitation of polaritons in two-dimensional (2D) materials[2,3] taking advantage of their reduced dimensionality and of the tunability of their dielectric properties via external stimuli such as temperature, strain[4] and charge carrier injection. Polaritons are quasi-particles arising from the coherent interaction of a charge polarization in a material and the e.m. field[5,6]. In the THz range, the charge polarization can consist of either oscillating free carriers in metals or doped-

semiconductors (e.g. graphene, black phosphorus) to form plasmon polaritons (PPs)[7–9], or vibrations of a polar crystal (e.g. hBN, Bi$_2$Se$_3$) to form phonon polaritons (PhPs)[10–12].

PPs in graphene exhibit high confinement factors ($\approx \lambda_0/40$ in the mid-infrared[8]) and long lifetimes[13]. Enhanced field confinement (up to $\approx \lambda_0/66$) has been obtained by exploiting a graphene-insulator-metal structure[3] hosting acoustic PPs, which result from the hybridization of the graphene PPs with their mirror image, induced in a metal plate[14]. Furthermore, the PPs' dispersion in graphene can be conveniently adjusted by electrical gating[8,15], exploiting the easy tunability of the Fermi level. However, a limitation of graphene PPs is represented by their isotropic propagation, which is not intrinsically suitable for directional control of light propagation and light-matter interaction[16],

[1]NEST, CNR - Istituto Nanoscienze and Scuola Normale Superiore, P. San Silvestro 12, 56127 Pisa, Italy. [2]Istituto di Fotonica e Nanotecnologie, Consiglio Nazionale delle Ricerche, Piazza Leonardo da Vinci 32, 20133 Milano, Italy. [3]School of Electronic and Electrical Engineering, University of Leeds, Leeds LS2 9JT, UK. ✉e-mail: evaariannaaurelia.pogna@cnr.it; miriam.vitiello@sns.it

and motivated the investigation of anisotropic 2D crystals such as α-MoO$_3$[17–19], α-GeS[20] and black phosphorus (bP)[21,22].

The orthorhombic layered lattice of $\alpha$-MoO$_3$, $\alpha$-GeS and bP is naturally endowed with large structural in-plane anisotropy, which results in large in-plane optical anisotropy that can result in an anisotropic polariton propagation. Light propagation in these media is described by a biaxial dielectric tensor $\boldsymbol{\varepsilon} = (\varepsilon_x, \varepsilon_y, \varepsilon_z)$ with distinct ($\varepsilon_x \neq \varepsilon_y \neq \varepsilon_z$) in-plane permittivities $\varepsilon_x$, $\varepsilon_y$ and out-of-plane permittivity $\varepsilon_z$. Elliptic in-plane polariton dispersion is expected in the case that the in-plane permittivities have the same sign, while hyperbolic in-plane dispersion is expected in the case that the signs are different. Anisotropic PhPs, with both elliptic and hyperbolic dispersions, have been observed in α-MoO$_3$ at mid-infrared and THz frequencies, and in GeS at THz frequencies. However, these modes are associated to the optical phonons of $\alpha$-MoO$_3$ and are hardly actively controlled. Recently, the active tuning of the surface PhPs resonance of 4H-SiC nanopillars[23] has been demonstrated via interband photoexcitation, exploiting the coupling of carrier plasma with optical phonons. However, the shift attained with a two time change of carrier density was just 1% and it was accompanied by a comparable line broadening. On the contrary, doped bP is expected to support anisotropic PPs, associated to anisotropic charge carrier oscillations, which can be actively and efficiently tuned by electrostatic gating[21,22,24].

Specifically, bP is a promising material for applications requiring a fine control of the device conduction[25], including a thickness-dependent direct band gap[26,27] (0.3 eV in the bulk[28] and 2 eV in the single-layer form[27,29,30]) and high room-temperature hole mobility (>1000 cm$^2$/Vs in the bulk[31] and >350 cm$^2$/Vs in 6 nm thin films[27]). Furthermore, the recent demonstration of controlled pulsed laser deposition of wafer-scale ultrathin bP (thicknesses from 0.53 nm to 12 nm)[32] has significantly enhanced its application potential for the development of devices suitable for the information industry[32].

The in-plane anisotropy of bP arises along the two main crystal axes on which the material is packaged, i.e. the armchair (AC) and zigzag (ZZ) directions, where the electrical and thermal transport are favored, respectively[33,34]. Quasi-particles excited in the material must then reflect its inherent anisotropy. Despite being a promising candidate for studying anisotropic polaritons of 2D crystals, the study of bP collective excitations is currently limited to theoretical investigations[21,22], and to experimental studies at mid-infrared[9] (with no directional sensitivity) and ultraviolet[35] frequencies. No experimental reports of PPs at THz frequencies exist, so far.

Here, we apply near-field photocurrent nanoscopy to investigate PPs in bP, in the THz frequency range, exploiting photo-thermoelectric (PTE) detection[3]. The anisotropy of the unveiled bP THz PPs is explored by mapping their propagation using field-effect transistor (FET) photodetectors with split gates. Specifically, we investigate the dependence of the PPs propagation on the crystallographic orientation, which is identified by polarization-resolved Raman[36]. The near-field photovoltage images reveal deep-subwavelength THz PPs with large confinement factor (up to $\lambda_0$/76 at 2.01 THz) that depends on the bP crystal orientation. We finally demonstrate the active tunability of the dispersion of the THz PPs via electrostatic control of the carrier density, providing the first demonstration of controlled nanoscale confinement of THz light, exploiting the anisotropic PPs of bP.

## Results and discussion
### Samples description
Single-crystalline ingots of bP are grown via a chemical vapor transport technique[37] and mechanical exfoliation by the standard adhesive tape technique is used to obtain thin flakes with thickness in the range 10–30 nm. Selected bP flakes, with thickness of 20 nm (sample 1, ~38 layers), 29 nm (sample 2, ~55 layers) and 15 nm (sample 3, ~28 layers), are encapsulated in hBN thin films by deterministic dry transfer[38] to prevent oxidation, which can otherwise

severely deteriorate the bP dielectric properties[39]. Unlike bP, hBN is an in-plane isotropic uniaxial dielectric ($\varepsilon_x = \varepsilon_y \neq \varepsilon_z$), with isotropic in-plane permittivities. Therefore, it is not expected to contribute to the in-plane anisotropy of the heterostructure. hBN is a polar crystal with regions of hyperbolicity associated to the formation of PhPs[40,41], which however fall in the mid-infrared range (780–830 cm$^{-1}$ and 1370–1610 cm$^{-1}$)[40] and do not contribute to the THz response of the heterostructure. hBN has been widely used for graphene encapsulation and gating[42], and generally results in an improved charge carrier mobility (by a factor 10 compared to SiO$_2$-supported graphene, $\mu > 25000$ cm$^2$/Vs)[43,44] and lower charge carrier density fluctuations (by a factor 3 compared to SiO$_2$ supported graphene, $n_0$ ~ $10^{10}$cm$^{-2}$)[43,44].

A set of FETs, with split back-gates (G$_1$, G$_2$) forming a p−n junction, are then fabricated by contacting a set of hBN/bP/hBN van der Waals heterostructures via a combination of electron beam lithography and metal deposition. Details on device fabrication and their stability towards degradation are reported in the Supplementary Information (Supplementary Note 1–3, Figs. 1, 11).

### Terahertz photocurrent nanoscoscopy: plasmon polaritons in black phosphorus
THz PPs are investigated by near-field photocurrent nanoscopy[3,45,46], using a commercial scattering-type scanning near-field optical microscope (s-SNOM, NeaSNOM from Neaspec/attocube), coupled to a set of monochromatic, single-plasmon waveguide THz quantum cascade lasers (QCLs), delivering 0.1–2 mW optical power at frequencies $\omega_0$ in the range 2.01–4.65 THz.

While the atomic force microscope (AFM) tip of the s-SNOM is raster scanning the sample, we simultaneously measure the sample topography and the photovoltage $\Delta V$ induced by the impinging THz beam, by reading the voltage at the source-drain FET electrical contacts[3,45,46], see Fig. 1a. The s-SNOM tip operates in tapping mode at frequency $\Omega$, such that the near-field component $\Delta V_n$ of source-drain voltage $\Delta V$, due to the local photoexcitation localized by the AFM tip[46], is isolated by lock-in detection at the harmonics $n = 1,2,..5$ of $\Omega$. Specifically, the second harmonic ($n = 2$) is chosen for the analysis of $\Delta V_n$ maps as discussed in the SI (Supplementary Note 4, Supplementary Figs. 13–14).

Figure 1b shows a near-field photovoltage $\Delta V_2$ map of device 1, embedding the heterostructure hBN (10 nm)/bP (20 nm)/hBN (15 nm), measured while applying a finite bias at the p-n junction (V$_{G1}$ = −2 V and V$_{G2}$ = 2 V) and illuminating the s-SNOM tip with a THz QCL emitting at 2.69 THz.

A finite near-field photovoltage $\Delta V_2$ is detected at the p-n junction, which we ascribe to PTE (see Supplementary Note 5, Supplementary Figs. S15 and S16). Since the energy gap of bulk bP ($E_g$ ~ 300 meV)[27] is much larger than the probed THz photon energy (<20 meV), the THz light is absorbed via intraband absorption, producing finite gradients of both electronic T$_e$ and lattice temperatures in the bP flake[46]. In the presence of a spatially varying Seebeck coefficient S$_e$ (associated with the carrier density gradient imposed by the different gate voltages, V$_{G1}$ ≠ V$_{G2}$), the T$_e$ gradient, caused by the local near-field THz photoexcitation, generates a net PTE photovoltage in the bP channel[3,45,47]. The change of photovoltage polarity is a signature of PTE response[48,49]. The far-field PTE voltage for a junction can be approximated as V$_{PTE}$ = $\Delta S_e \Delta T_e$, where $\Delta S_e$ is the variation of Seebeck coefficient at the junction and $\Delta T_e$ is the electronic temperature change compared to the lattice. Therefore, the V$_{PTE}$ signal amplitude depends non-trivially on the carrier density through S$_e$ (see Supplementary Note 2, Supplementary Fig. 7).

Interestingly, the $\Delta V_2$ maps reveal signal oscillations parallel to the junction line, which we attribute to the interference of THz PPs launched by the AFM tip of the s-SNOM and reflected at the bP flake edge, as recently observed in graphene[3]. The electric field pattern, generated

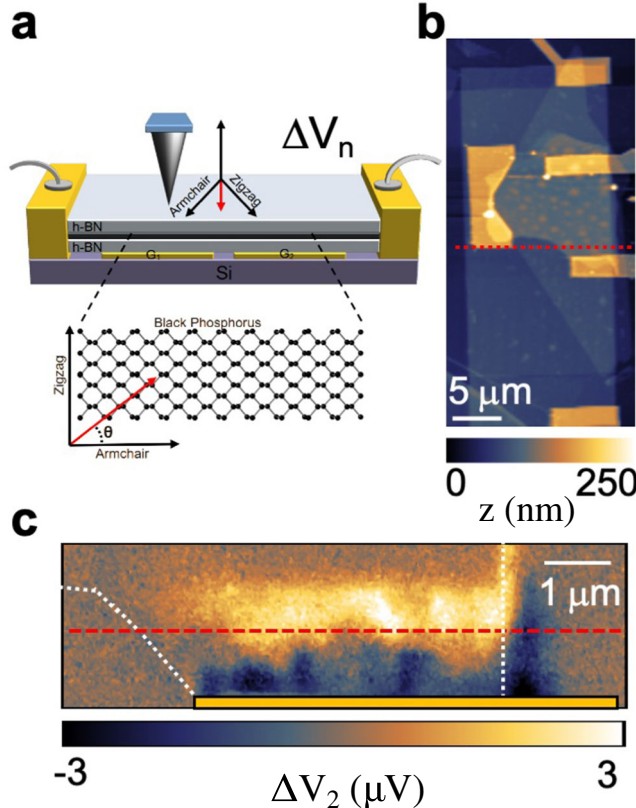

**Fig. 1 | Anisotropic THz-photocurrent nanoscopy of a black-phosphorus field effect transistor. a** Schematic diagram of the THz photocurrent nanoscopy experiments, based on the measurement of the near-field photovoltage $\Delta V_n$ induced in a black-phosphorus (bP) based heterostructure, by the laser-illuminated metal tip of an atomic force microscope (AFM) acting as near-field light source. The sample consists of a thin bP flake, encapsulated by hBN and transferred on a FET with a p-n junction defined by split back gates $G_1$ and $G_2$, used to control the carrier concentration. The near-field photovoltage $\Delta V_n$ for a given angle $\theta$ between the armchair (AC) crystal axis of bP and the junction line (red arrow). **b** AFM map in nm scale of device 1, with two gates forming a junction at the dashed red line between source and drain electrodes, which are used to monitor the photo-voltage. **c** Second harmonic near-field photovoltage $\Delta V_2$ map of device 1, embedding a hBN (10 nm)/bP (20 nm)/hBN (15 nm) heterostructure, illuminated with a THz quantum cascade laser (QCL) emitting at 2.69 THz, while applying gate voltages $V_{G1} = -2$ V and $V_{G2} = 2$ V at the junction. The bP flake profile is indicated by the white dotted lines, the electrode position by the yellow rectangle superimposed to the photovoltage map.

by the THz PPs propagation, spatially modulates the local energy dissipation and therefore $T_e$ and the PTE signal.

The near-field photovoltage $\Delta V_2$ is larger at the junction where the gradient in Seebeck coefficient is maximized and exponentially decreases while moving away from it, with a decay length of ~150 nm that is not limited by the tip resolution ($\delta y = 25$ nm, see Supplementary Note 4) and far-exceeds the size of gap between the two gates (~50 nm).

This decay can depend on several factors, including the carrier density profile induced by the gates, and the cooling length of the free-carriers. Since the PP wavelength depends on the carrier density, to analyze the oscillation periodicity, we cut the near-field photovoltage maps parallel to the junction line at a distance of 200 nm, mediating over a region of 50 nm orthogonal to the junction.

The orientation of the bP crystal with respect to the FET junction is identified by angle-resolved polarized micro-Raman spectroscopy[36] (see Methods and Supplementary Note 8, for details). In Fig. 2a, b, we show the intensity of the Raman fingerprints $A^2_g$ and $B_{2g}$ (normalized

to the intensity of the $A^1_g$ peak), as a function of the angle between the pump polarization and the junction line, for device 1 and device 2; device 2 embeds a hBN (6 nm)/bP (29 nm)/hBN (16 nm) heterostructure.

Since the $A^2_g/A^1_g$ ratio is expected to be maximized when the pump polarization is parallel to the ZZ axis[36], we retrieve the angle between the junction line and the AC direction, which is $\theta = 46°$ for device 1 and $\theta = 17°$ for device 2.

To verify that the observed $\Delta V_2$ modulations are due to PPs, we repeat the near-field photocurrent measurements at distinct excitation frequencies $\omega_O$. We retrieve the polariton in-plane momentum $k_p = 2\pi/\lambda_p$ at each $\omega = \omega_O$ from the periodicity $\lambda_p/2$ of the near-field photo-voltage $\Delta V_2$ oscillations, using a damped sinusoidal model after subtracting the background signal in the Fourier space (see Supplementary Note 9, Supplementary Figs. 25 and 26, and ref. 3). The error bar is determined as 95% confidence interval of the fit of the line profiles in the spatial coordinate.

Remarkably, the retrieved dispersions $\omega(k_p)$, in Fig. 2e, f, are $\theta$-angle dependent and compatible with that of THz PPs, which we model by computing the complex-value reflectivity $r_p$ for p-polarized light, considering the composition of the heterostructures and the crystal orientation of the bP (see Supplementary Note 6 Supplementary Figs. 17–20). This should be considered as an effective polariton dispersion along the propagation direction forming an angle $\theta$ with the AC direction. Due to the short propagation length of PPs along the ZZ direction[22], we expect the interferometric pattern at the junction, which is at >500 nm distance from the flake edges and FET electrodes, to be dominated by the propagation of the PPs along the AC direction, as recently shown in anisotropic $Ag_2Te$[50], for which a lower anisotropy is expected based on the ratio Rm of the effective masses along the two prominent in-plane directions $m_{eff,x}/m_{eff,y} = 2.3$ against $m_{eff,ZZ}/m_{eff,AC} = 7.7$ in bP.

The model includes the carrier density $n$ and the carrier mobility $\mu$ of bP as free parameters, and good agreement with the experimental dispersions is obtained using the values $\{n, \mu\} = \{1 \times 10^{19}\,cm^{-3}, 220\,cm^2/Vs\}$ for device 1, and $\{1 \times 10^{19}\,cm^{-3}, 178\,cm^2/Vs\}$ for device 2. Previous reports[51] on a bare 14 nm thick bP flake gave $n \sim 2.2 \times 10^{19}\,cm^{-3}$, $\mu \approx 380–540\,cm^2/Vs$. The model also includes the FET back gates, as 30-nm-thick gold layers, separated from the bP by the bottom hBN, in order to account for the PP hybridization[22] already observed in graphene-insulator-metal heterostructures[3]. The carrier density and the mobility play a major role in defining the PPs' dispersion (see Supplementary Note 6, Supplementary Fig. 20).

To corroborate our claim that the retrieved near-field photo-voltage line profiles are due to the propagation of tip-launched PP waves, and verify that the reflection from the electrodes is not altering the periodicity of the photovoltage oscillations at the p-n junction, we perform numerical simulations of the electric field distribution in the investigated devices (see Supplementary Note 10).

The retrieved PPs' dispersion is clearly different from that of acoustic PPs in graphene[3] as a consequence of the different electronic structure of bP. Unlike graphene, the low energy electronic structure of bP is not linear[29] and deviates from a parabolic trend for increasing film thicknesses[22], resulting into higher Drude weight and smaller effective mass. The inherent in-plane anisotropy is expected to lead to a propagation direction-dependent plasmon dispersion, with longer-living and higher frequency resonances along the AC direction, due to the smaller effective mass[52] and reduced damping, compared to the ZZ direction[22].

In both devices 1 and 2, PPs support strong THz field confinement, with a wavelength reduction at 2.01 THz of ~76 in device 1, and of ~73 in device 2, for which the junction is closer to the AC direction. These values are similar to the confinement factors found for graphene acoustic PPs[3], and larger than that reported in free-standing graphene[8,15].

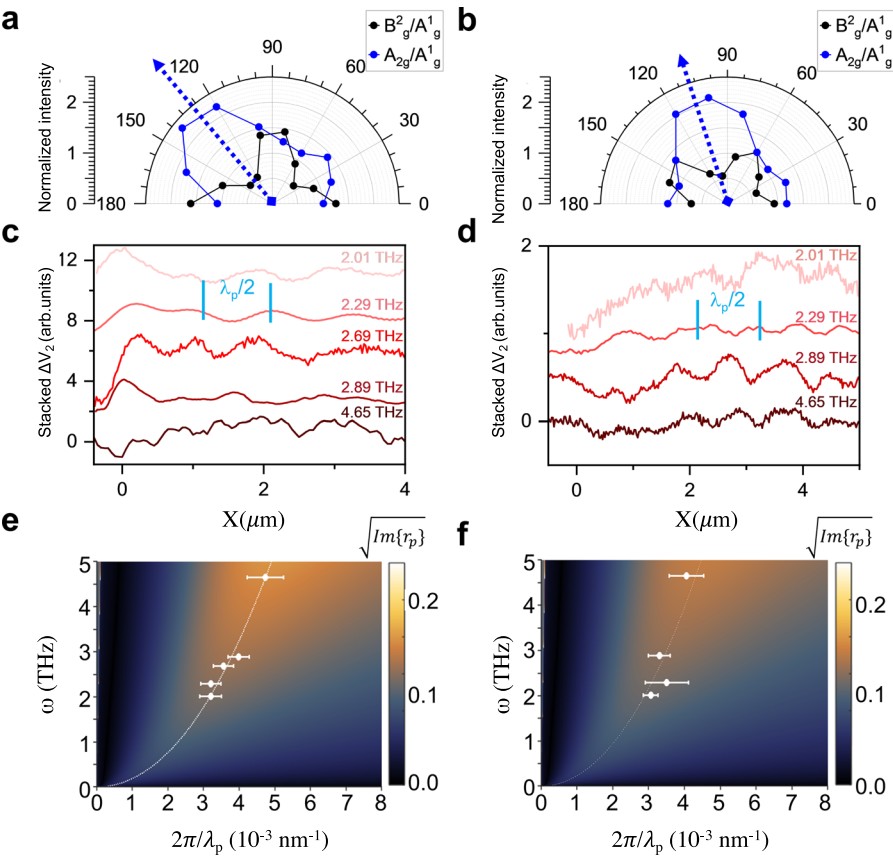

**Fig. 2 | Dispersion of anisotropic THz-plasmon polaritons in black phosphorus. a**, **b** Determination of the crystallographic orientation of a bP flake by polarization-dependent Raman spectroscopy. The polar plots show the intensity of $B_{2g}$ and $A^2_g$ Raman peaks of bP, normalized to the intensity of $A^1_g$ mode, as a function of the angle between the incident pump polarization and the junction line, for device 1 (**a**, hBN (10 nm)/bP (20 nm)/hBN (15 nm)) and device 2 (**b**, hBN (6 nm)/bP (29 nm)/hBN (16 nm)). The intensity of the $A^2_g$ peak is expected to be maximized when the pump polarization is aligned along the ZZ direction, indicated by the dashed blue arrow[62]. Near-field photovoltage profiles, as a function of the coordinate x, describing the distance from the flake edge (identified from the topography) along the junction line, in device 1 (**c**) and device 2 (**d**). Each profile is obtained by integrating the maps acquired using different THz QCLs over 50 nm across the junction

line. The periodicity $\lambda_p/2$ of the observed oscillations is used to extract the plasmon polariton (PP) momentum $k_p = 2\pi/\lambda_p$. The near-field photovoltage maps are acquired applying the gate voltages $V_{G1} = -2$ V and $V_{G2} = 2$ V (device 1) and $V_{G1} = -3$ V and $V_{G2} = 0$ V (device 2). Theoretical dispersion relation of PPs modes of device 1 (**e**) and device 2 (**f**), described by the imaginary part of the Fresnel reflection coefficient $r_p$ for p-polarized light (colored map), compared with the experimental dispersions (white dots) extracted from the line profiles in panels **c** and **d**; dashed white line is the maximum in the Im{$r_p$}. The error bars of the experimental points correspond to the 95% confidence interval extracted from the fit of the photovoltage line profiles. The theoretical dispersion is obtained by treating the carrier density $n$ and mobility $\mu$ of bP as fitting parameters to reproduce the data. The reported maps correspond to {$n, \mu$} = {$1 \times 10^{19}$ cm$^{-3}$, 220 cm$^2$/Vs} in **e**, and {$1 \times 10^{19}$ cm$^{-3}$, 178 cm$^2$/Vs} in **f**.

Such a large THz field confinement can be explained considering the out-of-phase charge oscillations in the bP flake and in the gate conducting plate that confines the plasmons vertical electric field within the dielectric gap provided by the bottom hBN layer of thickness $d$.

### Anisotropic propagation of THz plasmon polaritons

THz polariton dynamics (dispersion and damping) depend[22] on $d$, as well as on the thicknesses of all the layers in the heterostructure, and on the mobility $\mu$ and the doping $n$ of the bP sample. Therefore, to isolate the $\theta$-angle dependence of THz PPs dynamics, we engineer the four-gates ($G_{i,j}$ with i, j = 1, 2) FET sketched in Fig. 3a, with two orthogonal p-n junctions, and two drain contacts that enable the near-field photovoltage to be monitored simultaneously along two orthogonal directions in the same heterostructure. We fabricate device 3 embedding a hBN (10 nm)/bP (15 nm)/hBN (19 nm) heterostructure in the four-gates FET. The angle between the AC axis of the bP flake and the $G_{1j}$-$G_{2j}$ (j = 1,2) junction is $\theta = 5°$, as estimated by Raman spectroscopy (see Fig. 3b). Consequently, the two junctions of the FET are almost aligned to the principal crystal axes of the bP flake. The drain contacts are labeled $D_{AC}$ and $D_{ZZ}$ according to the crystallographic direction along which the photovoltage is measured. SEM images and electrical

characterization of device 3 are reported in the Supplementary Note 2 (Supplementary Figs. 5–6).

Whereas a finite near-field photovoltage emerges in the $D_{ZZ}$-S channel, with oscillations propagating along the AC direction (Fig. 3c), no PTE photovoltage is detected when we scan the junction line along the ZZ direction (Fig. 3d). This can be explained by the larger confinement and damping expected for THz PPs along the ZZ direction[22].

From the periodicity of the $\Delta V_2$ oscillation along the AC direction we extract the polariton wavelength ($\lambda_p = 1200$ nm at 2.01 THz, $\lambda_p/\lambda_o \sim 124$) and by repeating the measurement with different THz QCLs, we extract the dispersion of THz PPs in device 3 (see Supplementary Fig. 18).

THz anisotropic PPs (not acoustic) in the range 200–550 cm$^{-1}$ have been recently observed in WTe$_2$ nanodisks by far infrared reflectivity measurements[53]. Compared to WTe$_2$, bP is endowed by a much larger electronic anisotropy as testified by the effective mass ratio Rm[53], which is ~2 in WTe$_2$ and ~8 in bP. The stronger anisotropy of bP results into a stronger canalization of PPs along the more conductive direction, the AC direction[24]. Hyperbolic regimes, where the imaginary part of the dielectric permittivities along the in-plane directions have opposite sign, are predicted for both materials at frequencies approaching the interband absorption threshold along the

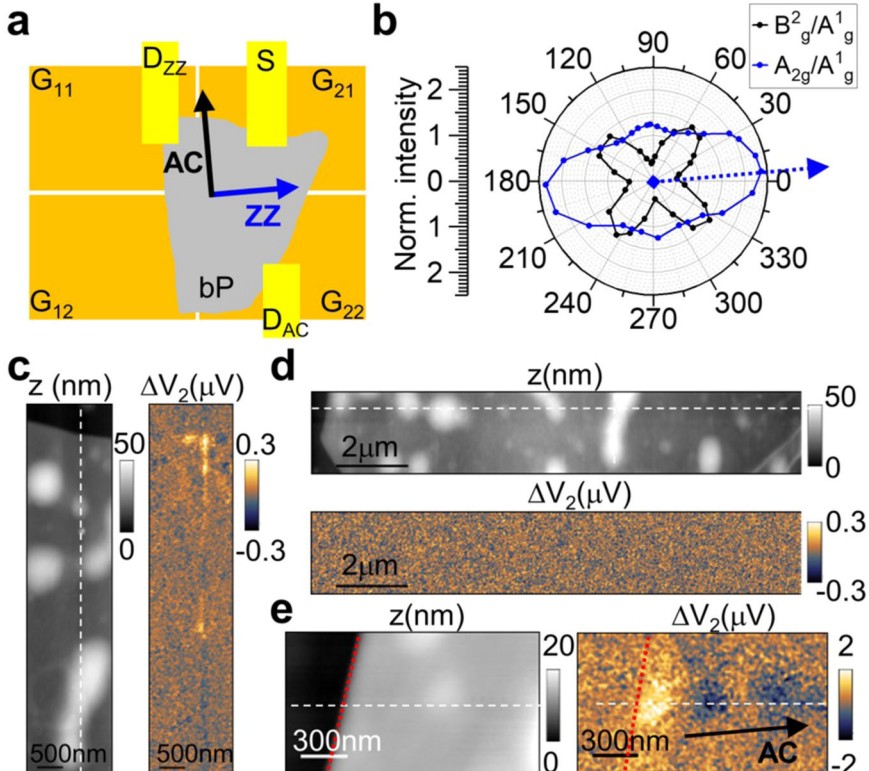

**Fig. 3 | Four-gates device for anisotropic THz photocurrent nanoscopy.**
**a** Sketch of the four-gates FET used for direction-sensitive photocurrent nano-scopy, exploiting the gates Gij (with i, j = 1,2) to polarize two p-n junctions along two orthogonal directions. The black and cyan arrows indicate the AC and ZZ directions of bP, as determined by Raman spectroscopy. The drain contacts $D_{AC}$ and $D_{ZZ}$ are used to monitor the photovoltage along the AC and ZZ directions, respectively.
**b** Normalized intensity of the Raman fingerprints of bP in device 3 as a function of the angle between the pump polarization and its long edge, used to determine the bP crystal orientation. The dotted cyan arrow indicates the ZZ direction. Topo-graphy z (gray scale) and near-field photovoltage $\Delta V_2$ (color scale) of device 3 along the junction line parallel to the AC direction (**c**) and ZZ direction (**d**), indicated by

the white dashed lines. The topography shows local deformations associated with debris of about 5–30 nm height. Finite $\Delta V_2$ is observed only at the junction line parallel to AC direction (**c**), while measuring at the $D_{zz}$- source (S) contacts. $\Delta V_2$ is measured while illuminating the sample with a THz-QCL emitting at 2.01 THz and applying gate voltages $V_{G11} = 0$ V and $V_{G21} = -0.6$ V in **c** and $V_{G22} = 0$ V and $V_{G21} = -0.6$ V in **d**. **e** Topography z (gray scale) and near-field photo-voltage $\Delta V_2$ (color scale) measured along the junction line parallel to AC direction (white dashed line), using the $D_{zz}$-S contacts, in a region of flat topography near a flake edge (red dotted lines). $\Delta V_2$ is measured while illuminating the sample with a THz-QCL emitting at 4.65 THz and applying gate voltages $V_{G11} = 0$ V and $V_{G21} = -0.2$ V. The black arrow indicates the AC direction along which we observe $\Delta V_2$ oscillations.

more conductive direction (AC in bP, b axis in WTe₂). However, for realistic doping levels (chemical potential >5 meV), the hyperbolic regime corresponds to mid-infrared frequencies[24,53] >12 THz in WTe₂ and >10 THz in bP, which we are not exploring here.

A number of atomically-thin in-plane anisotropic crystals with biaxial permittivity has been presently investigated, including ortho-rhombic crystals like bP, GeS, WTe₂, α-MoO₃, and triclinic dichalco-genides like ReSe₂. Amongst them, bP, GeS[54], WTe₂[55] show high carrier mobilities (of the order of $10^3$ cm²/Vs), while in nm-thick films of ReSe₂ and other typical TMDs (MoS₂ and WS₂) mobilities[56] are in the $10^2$ cm²/Vs range. However, the electronic and plasmonic anisotropy in GeS[54] and WTe₂[53] are quantified by a Rm -2 and in α-MoO₃ by a Rm-1.4[57], much lower than in bP. The studies of α-MoO₃ has therefore focused on the dielectric anisotropy associated to the optical phonon modes in the range 8–11 THz[2]. It's worth noticing that bP has a large thermoelectric power, resulting from the a Seebeck coefficient[25] (-300 μV/K), which is significantly (factor 100) larger than that of WTe₂[58], and comparable with that of ReS₂ (-900 μV/K)[56]. Moreover, large tunability of bP anisotropic optical response has been attained by strain and electric field application, anticipating promising route towards the development of reconfigurable nanophotonics devices. Relevantly for practical applications, the PPs dispersion is expected to show dependence on the charge carrier density $n$, which can be actively tuned by electrostatic gating[21]. The near-field photovoltage maps of device 3 as a function of the applied gate voltage are shown in

Fig. 4a, b. We observe a systematic increase of the polariton momen-tum $k_p$ with the gate voltage ($V_{G11}$). Since the bP flake is naturally p-doped, the carrier density is expected to increase as $V_{G11}$ is driven towards more negative values, as testified by the increase in the cur-rent measured along the $D_{ZZ}$-S channel (see Supplementary Note 2, Supplementary Fig. 6). The decrease in the polariton wavelength cor-responds to an increase in the THz field confinement.

Active control of PPs in bP can be obtained via electrical control of the carrier density, or by all-optical control of the charge carrier dis-tribution, as recently demonstrated at mid-infrared frequencies[9], exploiting the ultrafast optical doping induced by the interband absorption of a near-infrared pulse. Similarly, PPs at THz frequencies could be controlled optically, opening intriguing perspectives for the development of active optical switches for electronic waves[9]. More-over, the orthorhombic structure of bP has highly anisotropic elastic properties[59] and strong variation of the electronic properties are expected upon strain engineering[4], which therefore represents an additional control mechanism for tuning the conductivity and PP excitations of bP.

In conclusion, we have investigated THz polaritons of hBN-encapsulated bP, providing the experimental observation of strongly confined PPs with anisotropic propagation. The real-space mapping of the local THz-induced PTE near-field response in a bP-FET reveals strongly enhanced field confinement and a selective propagation along the AC direction, in which longer polariton wavelengths and

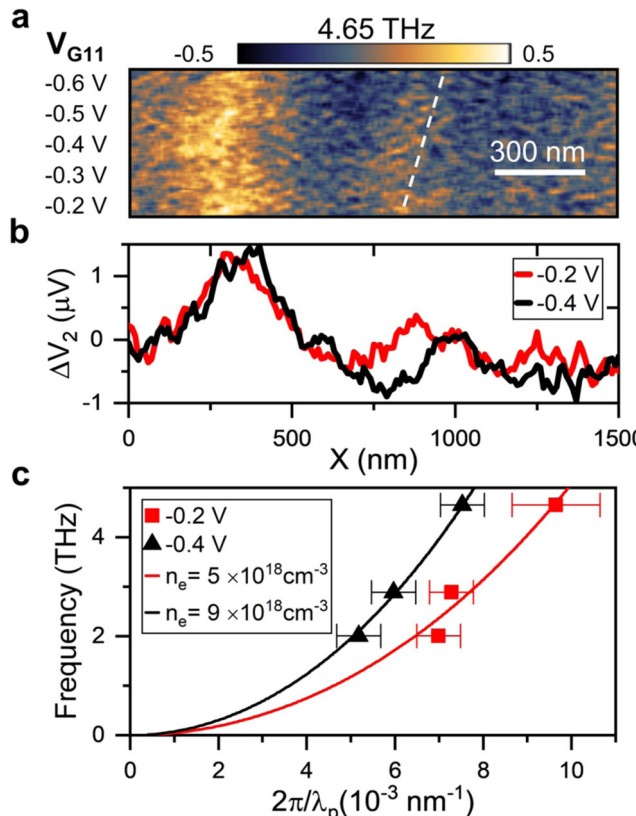

**Fig. 4 | Gate tunability of the THz PP wavelength. a, b** Near-field photovoltage fringes $\Delta V_2$ along the AC direction measured at 4.65 THz (**a**) while varying the gate voltage $V_{GII}$. The white dashed line follows the second maximum position. The slow varying component observed in near-field maps is removed with Fourier transform filtering as in ref. 3. **b** Linear profiles extracted from the near-field photovoltage maps in panel **a** acquired at 4.65 THz with $V_{GII}$ equal to −0.2 V (red line) and −0.4 V (black line). **c** PP dispersion measured while applying $V_{GII}$ equal to −0.2 V (red dot) and −0.4 V (black dot) as compared with the dispersion evaluated as the maximum of Im{$r_p$} for $\mu = 160$ cm²/Vs and $n = 5 \times 10^{18}$ cm⁻³ (red line) and $9 \times 10^{18}$ cm⁻³ (black line). The error bars of the experimental points correspond to the standard deviation extracted from the fit of the photovoltage line profiles.

reduced damping is expected. The experimental dispersions measured in three devices along the FET junction line, forming an angle 5° $< \theta <$ 46° with the AC axis of bP, provide a benchmark to the theoretical modeling of bP plasmon polaritons. We reproduced the polariton dispersion by computing the losses of the device reflectivity by using the transfer matrix method to account for the multilayer composition of the heterostructures. Remarkably, we detect a variation in the PP dispersion, even for small variation of the carrier density (up to 20%), reflecting the large tunability by electrostatic gating. Additional tunability can be obtained by controlling the thickness of the bottom hBN that defines the gap between bP and the conducting plate.

The four-gates FET geometry allows for direction-resolved PTE detection of PP propagation and can be applied for investigating the in-plane anisotropy of alternative 2D materials, overcoming the intrinsic limitation of the s-SNOM techniques, which preferably couple out-of-plane field polarization. The anisotropy of the dielectric response of bP can be exploited for engineering THz hybrid polaritons in van der Waals heterostructures. In the mid-infrared, hBN/bP heterostructures has been exploited for tailoring hybrid polariton modes based on the PhPs of hBN, inheriting in-plane anisotropy from the bP substrate[60]. The same idea can be extended to the THz range, exploiting hybridization, given the fulfillment of phase matching conditions[61].

The photoelectrical detection of PPs directly suggests possible on-chip functionalities based on the exploitation of active-tunable PPs. Our findings can help to leverage the extreme directional confinement provided by PPs in bP and exploit the material optical anisotropy to develop optoelectronic devices such as metasurface biosensors, optical modulators, molecular trapping, and photodetectors, exploiting the sensitivity of polaritons to the environment, and the strong enhancement of light-matter interaction achieved by light confinement to the nanoscale.

## Methods
### Angle-dependent Raman measurements
Micro-Raman spectra are collected using a commercial Raman spectrometer (Horiba Xplora) using a continuous wave (CW) 638 nm laser with 0.3 mW incident power, 1 μm spot radius, averaging 40 acquisitions lasting 0.5 s each, and dispersing the scattered intensity with a 1800gr/mm grating. The pump laser polarization is linear and the reciprocal orientation between the pump polarization and the sample is controlled by placing the sample on a continuous rotation mount with 1 degree of sensitivity. The laser spot position on the sample is kept constant during the measurement. The measurements are performed in air, in complete darkness and last roughly 1 h per heterostructure. We analyze the angle dependent intensity of the Raman peaks $A^1_g$, $A^2_g$ and $B_{2g}$ modes at frequencies $\nu_{AIg} = 366$ cm⁻¹, $\nu_{A2g} = 470$ cm⁻¹ and $\nu_{B2g} = 442$ cm⁻¹. In order to determine the crystallographic direction of the bP flakes, we take into account that a distinct trend for the intensity of the peaks is expected depending on the film thickness and pump laser wavelength[36]. The hBN/bP/hBN heterostructures embedded into devices 1 and 2 have been characterized prior to the near-field experiment, by focusing the pump spot on a flake region outside of the photoconductive channel. The heterostructure embedded into device 3 has been characterized by Raman spectroscopy after the near-field experiment.

### Photocurrent nanoscopy
Near-field photovoltage measurements are performed using a commercial NeaSNOM (Neaspec/attocube) coupled with custom-made single plasmon waveguide THz-QCL[46]. The collimated THz beam from the THz-QCL is focused on the tip-sample system by a 25 mm focal length off-axis parabolic mirror integrated into the NeaSNOM. We use commercial AFM tips (25PtIr300B-H, Rocky Mountain Nanotechnology), with 20 nm apex radius, operated at tapping frequency Ω-70–110 kHz and amplitude A - 100–150 nm to acquire near-field maps. While we raster-scan the sample, we simultaneously monitor the sample topography and the electrical readouts of the FETs. During the measurements, we apply no bias to the FET channel to reduce the electrical noise and we keep the source contact and the AFM tip at the same ground as the NeaSNOM. Since we are not applying a bias to the FET channel, we are not sensitive to the bolometric response of bP[48]. The photovoltage between the source and drain contacts is preamplified with a high-input-impedance (R = 600 MΩ), low-noise voltage amplifier (Model 1204 DL Instruments) and then analyzed with the acquisition card of the s-SNOM that filters the components at the harmonics of the tip tapping frequency. Near-field microscopy characterization is performed in a dark, inert atmosphere purged with N₂ to keep humidity levels <3%, to reduce degradation risk.

## Data availability
The data presented in this study are available on request from the corresponding authors.

## Code availability
The relevant computer codes supporting this study are available from the authors upon request.

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

## Acknowledgements

This work was supported by the European Research Council through the ERC Consolidator Grant (681379) SPRINT (M.S.V.), the European Research Council through the ERC Proof of Concept Grant (101081567) STAR (M.S.V.), the Graphene Flagship (core 3) (M.S.V.), and the EPSRC (UK) programme grants HyperTerahertz (EP/P021859/1) (E.H.L. and G.A.D.) and TeraCom (EP/W028921/1) (E.H.L. and G.A.D.).

## Author contributions

The authors confirm contribution to the paper as follows: study conception and design: E.A.A.P., L.V. and M.S.V.; design and fabrication of the devices: L.V.; growth of the THz QCLs: L.L., E.H.L., G.A.D.; near-field measurements: E.A.A.P. and V.P. Data analysis: E.A.A.P.; Interpretation of results: E.A.A.P., L.V., M.S.V.; manuscript preparation: E.A.A.P., M.S.V with contribution from L.V. and V.P; coordination and supervision of the project: M.S.V. All authors reviewed the results and approved the final version of the manuscript.

## Competing interests

The authors declare no competing interests.
