## [Peer Review File · Nature Communications]

Near-field detection of gate-tunable anisotropic plasmon polaritons in black phosphorus at terahertz frequenciesEditorial Note: This manuscript has been previously reviewed at another journal that is not operating a transparent peer review scheme. This document only contains reviewer comments and rebuttal letters for versions considered at *Nature Communications*.

REVIEWER COMMENTS

Reviewer #6 (Remarks to the Author):

Pogna et al. report the near-field detection of terahertz anisotropic plasmon polaritons using photocurrent nanoscopy. I believe that this result is significant and novel enough to warrant publication in a reputable journal, including Nature Communications. However, I feel that the theoretical/numerical analysis of the measured near-field photovoltage maps must be supplemented to convince the readers in the near-field community.

1. As noted by Reviewer 1 (comment #2) and Reviewer 3 (comment #3), I also believe that a detailed description of the fitting method is essential, especially since the near-field interference signals presented in Figure 2 may not be sufficiently clear for extracting the polariton wavelength without ambiguity. Although the authors have mentioned following the description in Ref. 3, it is advisable to provide more comprehensive information. For instance, I recommend that the authors include the Fourier transformation of the measured interference patterns and a comparison between the fitting curves and the measured real-space signals, as demonstrated in Supplementary Section 4 of Ref. 3.

2. As suggested by Reviewer 4 (comment #4 and #5), since the samples contain many objects (source/drain electrodes, gate electrodes, and the material edges) that could scatter/excite surface polaritons, a full-wave simulation of the device is needed to examine their influence on the near-field signal. Although the authors provided a rudimentary interference pattern generated by the simple superposition of two polariton modes in response to Reviewer 4's comment, I do not think this is sufficient.

Answer to the referee's questions and list of changes
NCOMMS-23-49445-T

Reviewer #6 (Remarks to the Author):

Pogna et al. report the near-field detection of terahertz anisotropic plasmon polaritons using photocurrent nanoscopy. I believe that this result is significant and novel enough to warrant publication in a reputable journal, including Nature Communications.

However, I feel that the theoretical/numerical analysis of the measured near-field photovoltage maps must be supplemented to convince the readers in the near-field community.

ANSWER

We thank the reviewer for having dedicated time to the careful reading of our manuscript and for considering our work *significant and novel enough to warrant publication in a reputable journal, including Nature Communications*. We have addressed his/her comments and we hope that he/she now found the manuscript acceptable for publication.

1. As noted by Reviewer 1 (comment #2) and Reviewer 3 (comment #3), I also believe that a detailed description of the fitting method is essential, especially since the near-field interference signals presented in Figure 2 may not be sufficiently clear for extracting the polariton wavelength without ambiguity. Although the authors have mentioned following the description in Ref. 3, it is advisable to provide more comprehensive information. For instance, I recommend that the authors include the Fourier transformation of the measured interference patterns and a comparison between the fitting curves and the measured real-space signals, as demonstrated in Supplementary Section 4 of Ref. 3.

ANSWER

We thank the reviewer for pointing this out. Following this remark, we have included the description of the fitting procedure in the main text.

Specifically:

-We have included a detailed description of the analysis of the polariton interference patterns in the Supplementary Information (SI). We here report the Fourier transforms of the line profiles plotted in Figure 2 and the fitting curves of the real-space signals after background subtraction. We have also included the fitting of the line-profiles used to obtain the dispersion curve of device 3 shown in Fig. 4, in the main text.

The added section is reported below:

Supplementary Information, Page 22 line 10

9. Measurement of the bP polaritons wavelength

9.1 Analysis of the near-field photovoltage profiles of Figure 2

The momentum k_p and the wavelength λ_p of the THz polaritons of bP are extracted from the periodicity of the near-field photovoltage ΔV_2 line profiles perpendicular to the pn-junctions. The underneath assumption

is that the polariton waves are launched by the *s*-SNOM tip and reflected at the flake edges, so that the period of the ΔV_2 oscillations corresponds to half of the polariton wavelength, $\lambda_p/2$. As previously reported in graphene,¹ the PTE near-field signal of bP contains a background signal arising either from the non-plasmonic heating of the charge carriers or from local plasmonic heating. In both cases such a signal does not contribute to the oscillating component of the signal due to the propagating plasmon polaritons. To isolate the latter contribution, we subtract the signal background before fitting the photovoltage profiles with a damped sinusoidal function. To this aim, we evaluate the Fourier transform (FT) of the line profiles of Figure 2, using zero padding and triangular apodization. The moduli of the FT of the line profiles of Figure 2c, 2d are reported in Figure S25a and Figure S26a, respectively. The background is subtracted, setting a high-pass filter, corresponding to the shaded areas in Figure S25 and S26. The peaks attributed to the propagating polaritons are marked with red dotted lines for an easier comparison with the dispersion curves shown in Figure 2.

Figure S25b and Figure S26b plot the near-field photovoltage profiles after background subtraction together with the fit of the real-space line profiles used to extract the oscillation periodicity.

Figure S25 | Fitting of the near-field photovoltage line profiles of device 1. a) Moduli (black solid lines) of the Fourier transform (FT) of the near-field photovoltage profiles of device 1, reported in Figure 2c, measured at different frequencies from 2.01 THz (bottom panel) to 4.65 THz (upper panel). The FTs are shown as a function of the polariton momentum q (the spatial frequency multiplied by two). The curves are normalized to the maximum. Grey shaded areas indicate the frequency ranges filtered out to subtract the background due to local heating. Red dotted lines identify the position of the peak attributed to the propagating polaritons. b) Near-field photovoltage profiles after the background subtraction (black dots), as a function of the spatial coordinate, together with the damped sinusoidal fitting function (red solid line) used to extract the polariton momentum and the dispersion curve of device 1 in Figure 2e.

Figure S26 / Fitting of the near-field photovoltage line profiles of device 2. **a)** Moduli (black solid lines) of the Fourier transform (FT) of the near-field photovoltage profiles of device 2, reported in Figure 2d, measured at different frequencies from 2.01 THz (bottom panel) to 4.65 THz (upper panel). The FT are shown as a function of the polariton momentum q (the spatial frequency multiplied by two). The curves are normalized to the maximum. Grey shaded areas indicate the frequency ranges filtered out to subtract the background due to local heating. Red dotted lines identify the position of the peak attributed to the propagating polaritons. **b)** Near-field photovoltage profiles after the background subtraction (black dots), as a function of the spatial coordinate, together with the damped sinusoidal fitting function (red solid line) used to extract the polariton momentum and the dispersion of device 1 in Figure 2f.

It is worth mentioning that the attenuation of the PTE signal can be attributed to several factors, including the temperature profile, that depends on the plasmon induced heating, and the Seebeck coefficient that depends on the carrier density profile induced by electrostatic gating.

9.2 Analysis of the near-field photovoltage profiles of Figure 4

Figure S27 shows the near-field photovoltage ΔV_2 profiles measured in device 3 along the AC direction, while applying a finite gate voltage $V_{G11} = -0.4, -0.2$ V, together with the damped sinusoidal fitting functions, adopted to extract the dispersion characteristic of Figure 4c, main text.

Figure S27 | Fitting of the near-field photovoltage line profiles of device 3. Spatial profile of the near-field photovoltage after the background subtraction (black dots), plotted together with the damped sinusoidal fitting function (red solid line) used to extract the polariton momentum and the dispersion characteristic of device 1, reported in Figure 4c (main text), while applying a finite gate voltage $V_{G11} = -0.2$ V (left panels), and $V_{G11} = -0.4$ V (right panels).

- We add the error bar to the dispersion points in Figure 4c.
- We have also modified the main text as follows:

Page 10, line 5

We retrieve the polariton in-plane momentum $k_p = 2\pi/\lambda_p$ at each $\omega = \omega_0$ from the periodicity $\lambda_p/2$ of the near-field photovoltage ΔV_2 oscillations, using a damped sinusoidal model after subtracting the background signal in the Fourier space (see SI section 9, Fig. S25-S26, and Ref. 3). The error bar is determined as 95% confidence interval of the fit of the line profiles in the spatial coordinate. “

2. As suggested by Reviewer 4 (comment #4 and #5), since the samples contain many objects (source/drain electrodes, gate electrodes, and the material edges) that could scatter/excite surface polaritons, a full-wave simulation of the device is needed to examine their influence on the near-field signal. Although the authors provided a rudimentary interference pattern generated by the simple superposition of two polariton modes in response to Reviewer 4's comment, I do not think this is sufficient.

ANSWER

Following the referee's remark, we have developed a full-wave simulation of the electric distribution in our devices, based on the finite element method, as requested by Reviewers 4 and 6, to verify that the reflection of the polariton wave at the electrodes does not affect the correct retrieval of the polariton momentum.

The simulation shows that the presence of the electrodes, which in the three investigated devices are placed at a distance $> 2 \mu\text{m}$ from the p - n junction in which we measure the PTE signal, is not altering the periodicity of the measured interference pattern. Therefore, the photovoltage profiles can be safely analyzed to retrieve the polariton dispersion. In addition, the simulation shows that edge modes are not contributing to the signal at the p - n junction.

In conclusion the measured dispersion is compatible with tip launched modes as we correctly assume in the manuscript.

We have modified the manuscript as follows:

Page 11, line 7

We added the text:

To corroborate our claim that the retrieved near-field photovoltage line profiles are due to the propagation of tip-launched plasmon polariton waves and verify that the reflection from the electrodes is not altering the periodicity of the photovoltage oscillations at the p - n junction, we perform numerical simulations of the electric field distribution in the investigated devices (see SI, section 10)

We have inserted a dedicated section in the Supplementary Information:

Page 25, line 8

10. Numerical simulations of s -SNOM polariton interferometric patterns

We perform a full-wave simulation (Comsol Multiphysics) of the electric field distribution in the investigated bP devices, including the actual flake size and electrodes distances. The s -SNOM tip illuminated by the THz QCLs is modelled as a vertically oriented dipole source placed above the device surface at a distance $z = 150 \text{ nm}$, as conventionally done for predicting the near-field of an elongated AFM tips illuminated by p -polarized light [21-23].

We calculate the component of the electric field perpendicular to the sample $E_z(x,y)$, at a distance $z = 50 \text{ nm}$ from the surface, as a function of the dipole position in the xy plane, see Figure S28.

Figure S28 | Simulation of the near-field distribution of polaritons in device 1 at 2 THz frequency. **a-c**, Real part of the out-of-plane component of the electric field $\text{Re}\{E_z\}$ at $z=50$ nm, as generated by a vertically oriented electric dipole placed at $z=150$ nm in different x - y positions: at the center of the transistor channel (a), at 1.2 μm from the bottom electrode (b), at 600 nm from the bottom electrode. **d-f**, Electric field (black solid line) along the black dashed line \mathbf{u} perpendicular to the junction (parallel to the electrodes) extracted from the map in panel a (d), panel b (e) and panel c (f), together with the fit (red solid line) with the function $\text{Re}\{A \exp(iqX)/\sqrt{|X|}\}$, where q is the complex-valued polariton momentum such that $k_p = \text{Re}\{q\}$. **g-i**, Electric field (black solid line) along the blue dashed line \mathbf{v} parallel to the junction (perpendicular to the electrodes) extracted from the map in panel a (g), panel b (h) and panel c (i), together with the fit (red solid line) with the function $\text{Re}\{A \exp(iqY)/\sqrt{|Y|}\}$, where q is the complex-valued polariton momentum. The \mathbf{u} direction forms an angle of -17° with the zigzag direction of bP which is taken along the x axis.

The multilayer structure of our devices is taken into account including 8 different material domains: the silicon substrate (thickness 100 μm , isotropic relative permittivity 6.3), a gold layer describing the back gates (thickness 30 nm, refractive index $n=1380+i1450$), the bottom and top hBN layers (thickness 19 and 10 nm, refractive index $2.2+0.005i$ (in-plane direction) and $2.05+0.005i$ (out of plane direction)), the bP flake, two gold source-drain electrodes (thickness 45 nm, same refractive index as the gate), the air surrounding the device, isotropic relative permittivity 1). The bP flake is described by a diagonalized dielectric tensor using the complex-value dielectric functions along the zigzag (x) armchair (y) and stacking (z) directions, used for the simulation of the reflectivity of device 1, as shown in Figure S20. To model device 1, we introduce a bP flake of thickness $d = 10$ nm, rectangular lateral size 7 $\mu\text{m} \times 13$ μm , rotated by 17° with respect to the zigzag axis. We placed the electrodes at a relative distance of 6.3 μm to evaluate

the impact of the interferometric pattern generated by propagating polaritons that we measure at the p-n junction by PTE detection.

Figure S28 shows the electric field distribution evaluated with the dipole at the center of the transistor channel or near the bottom electrode at distances 1.2 μm and 600 nm, respectively. The elliptic wavefront of generated waves, reported in the panels a-c, proves the anisotropy of the bP dielectric tensor. We then analyze (S27d-i) the E_z line profiles along two orthogonal directions, parallel \mathbf{v} and perpendicular \mathbf{u} to the p-n junction, which form angles of -17° and 73° with the zigzag direction, respectively.

The traces measured perpendicular to the p-n junction are fitted with a radially propagating damped wave $A \exp(iq \cdot X) / \sqrt{X}$ to retrieve the polariton momentum $k_p = \text{Re}\{q\}$ at 2 THz, which is compatible with the experimentally determined momenta. Interestingly the retrieved momentum is barely affected both by the position of the dipole and its proximity to the electrodes. The polariton momentum extracted when the dipole is approached to the electrode, at a distance of 600 nm, is equal to that obtained with the dipole at the center of the channel within the uncertainty of photovoltage measurement ($>0.3 \mu\text{m}^{-1}$).

Conversely, the polariton reflection at the electrodes affects the line-profiles along the \mathbf{v} direction, parallel to the p-n junction, producing a modification of the retrieved momentum of $0.6 \mu\text{m}^{-1}$, when comparing the fringes measured with the dipole at the center and the dipole at 600 nm from the electrode. It is worth mentioning that, in our experiments for all devices, we measure the photovoltage oscillation at a p-n junction located at a distance larger than $2 \mu\text{m}$ from the electrodes, meaning that the measured momentum represents a reliable estimation of the effective polariton momentum.

The simulations of Fig. S28 confirm that the experimental s-SNOM line profiles reveal a polariton mode that: (i) is launched by the tip (not by the edges); (ii) the reflection of the polariton wave from the electrodes does not affect the field distribution at the junction perpendicular to the probed p-n junction.

Also, these conclusions, valid for sample 1, even more safely apply to device 2 and device 3 including bP flakes of larger size and electrodes located at larger distances from the p-n junction.”

We added the following references to the SI

[21] Chen, S., Bylinkin, A., Wang, Z. et al. Real-space nanoimaging of THz polaritons in the topological insulator Bi₂Se₃. Nat Commun 13, 1374 (2022). <https://doi.org/10.1038/s41467-022-28791-x>

[22] Nikitin, A. Y. et al. Real-space mapping of tailored sheet and edge plasmons in graphene nanoresonators. Nat. Photon. 10, 239–243 (2016).

[23] Ma, W. et al. In-plane anisotropic and ultra-low-loss polaritons in a natural van der Waals crystal. Nature 562, 557–562 (2018).

[24] Optical constants from the far infrared to the x-ray region: Mg, Al, Cu, Ag, Au, Bi, C, and Al₂O₃. H.-J. Hagemann, W. Gudat, and C. Kunz. J. Opt. Soc. Am. 65, pp. 742-744 (1975).

The Section numbers in the Supplementary Information file have been renumbered, accordingly.

REVIEWERS' COMMENTS

Reviewer #6 (Remarks to the Author):

The authors have satisfactorily addressed all my concerns. I am now delighted to support the publication of this work.